# Targeted Therapy and Immunotherapy for Heterogeneous Breast Cancer

**DOI:** 10.3390/cancers14215456

**Published:** 2022-11-06

**Authors:** Xiaolu Sun, Kuai Liu, Shuli Lu, Weina He, Zixiu Du

**Affiliations:** 1Engineering Research Center of Cell & Therapeutic Antibody, Ministry of Education, School of Pharmacy, Shanghai Jiao Tong University, 800 Dongchuan Road, Shanghai 200240, China; 2Shanghai Jiao Tong University School of Medicine, 227 South Chongqing Road, Shanghai 200025, China

**Keywords:** heterogeneous breast cancer, targeted therapy, immunotherapy, targeted drug delivery systems

## Abstract

**Simple Summary:**

Breast cancer (BC) is a common malignancy with molecular diversity, i.e., heterogeneity. Aside from routine clinical treatments, such as chemotherapy and radiotherapy, which have side effects and tumor resistance, troubling patients and doctors, targeted therapy based on molecular classifications and immunotherapy with novel approaches to reprogram the immune system offer solutions to improve prognosis, anti-tumor efficacy, and address drug resistance. Here, we review the wide range of molecular classifications of heterogeneous BC, emphasize targeted therapy and immunotherapy, and provide insights into the significance of targeted drug delivery systems.

**Abstract:**

Breast cancer (BC) is the most common malignancy in women worldwide, and it is a molecularly diverse disease. Heterogeneity can be observed in a wide range of cell types with varying morphologies and behaviors. Molecular classifications are broadly used in clinical diagnosis, including estrogen receptor (ER), progesterone receptor (PR), human epidermal growth factor receptor 2 (HER2), epidermal growth factor receptor (EGFR), vascular endothelial growth factor receptor (VEGFR), and breast cancer gene (BRCA) mutations, as indicators of tumor heterogeneity. Treatment strategies differ according to the molecular subtype. Besides the traditional treatments, such as hormone (endocrine) therapy, radiotherapy, and chemotherapy, innovative approaches have accelerated BC treatments, which contain targeted therapies and immunotherapy. Among them, monoclonal antibodies, small-molecule inhibitors and antibody–drug conjugates, and targeted delivery systems are promising armamentarium for breast cancer, while checkpoint inhibitors, CAR T cell therapy, cancer vaccines, and tumor-microenvironment-targeted therapy provide a more comprehensive understanding of breast cancer and could assist in developing new therapeutic strategies.

## 1. Introduction

Breast cancer (BC) is the most common female malignancy, with an estimated 2.3 million new cases each year. Female breast cancer has already surpassed lung cancer as the most commonly diagnosed cancer in 2020 [1] and is the leading cause of tumor-related mortality in women worldwide.

Due to the high heterogeneity of breast cancer, establishing precise prevention and treatment programs remains a challenge. Heterogeneity can be observed in a variety of cell types with different morphologies and behaviors, and these differences have served as the foundation for disease classification, eventually dividing it into two categories: intertumor heterogeneity and intratumor heterogeneity [2]. The former can be observed in a variety of patients, whereas the staging system and histopathological classification are the most effective methods for reflecting clinical diagnosis. The latter manifests within a single tumor and can be reflected at the genomic, transcriptomic, and proteomic levels of expression, posing diagnostic and therapeutic challenges. To comprehend and address the challenges posed by heterogeneity, molecular classification is an area of study that requires immediate attention.

For intertumor or intratumor, the existence of heterogeneity doubtlessly increases the risk of the mutation being expressed in the genetic features of the cells. No matter what kind of heterogeneity cancer has, cells will inevitably be affected by a series of mutation events and acquire the ability to evade the immune system and overcome anti-tumor host defenses.

Despite advances in molecular biomarker knowledge, there has been little progress in overcoming this malignant disease overall under the current clinical guidelines. It is worth noting that, in addition to traditional treatments, such as hormone therapy, chemotherapy, and surgical treatment, various effective treatment strategies, such as targeted therapy and immunotherapy, have been developed.

In a nutshell, this review will conclude the molecular classification of heterogeneous breast cancer, highlight the current treatment hotspots, with an emphasis on targeted therapy and immunotherapy, and provide insights into the significance of targeted drug delivery systems.

## 2. Molecular Classification

Molecular classifications are widely used in clinical diagnosis and serve as indicators of tumor heterogeneity, allowing patients to be risk-stratified for subsequent personalized therapy. Estrogen receptor (ER), progesterone receptor (PR), and human epidermal growth factor receptor 2 (HER2) are examples of classical biomarkers that have been routinely selected for pathology experiments with well-established staining protocols all over the world, while other biomarkers, BRCA, EGF, and VEGF, etc., have already been identified and have the potential to be chosen for precise detection and treatments (Figure 1).

### 2.1. ER and PR

In the 1970s, the ER biomarker was first recognized and utilized to estimate the prognosis and indicate early recurrence. As the single predictive factor identified in BC and the indicator of endocrine therapy, the ER biomarker is most powerful in treatments with approximately 50% effective response to anti-estrogen or aromatase inhibitors for the ER-positive (ER+) phenotype, which comprises up to 70–80% of BC subtypes [3]. Meanwhile, the PR biomarker, induced by endocrine, is also a co-indicator of endocrine therapy. The PR-positive (PR+) phenotype always comes with ER+ and consists of 55–65% of BC. Compared with other classified subtypes, ER/PR−, ER−/PR−, or ER+/PR−, ER/PR+ tumors are the most responsive to endocrine therapy due to the characteristics of hormone-dependent growth retained by the tumor cells.

### 2.2. HER2

However, when it comes to the ER/PR− phenotype, including the HER2+ phenotype, keywords are more about the problems of low malignancy, poor prognosis, and high recurrence probability, troubling both doctors and patients. Several studies have revealed that HER2 biomarkers are tightly associated with poor prognosis and the others aforementioned [4,5]. HER2, a transmembrane tyrosine kinase receptor, regulates proliferation, cell survival, and adhesion through activation of various downstream signaling pathways, such as rat sarcoma virus/rapidly accelerated fibrosarcoma/mitogen-activated protein kinase kinase/extracellular-signal-regulated kinase (Ras/Raf/MEK/ERK), and phosphoinositide 3-kinase/protein kinase B/mammalian target of rapamycin (PI3K/AKT/mTOR) [6]. Therefore, the emergence of HER2 biomarkers has alleviated the difficulty in treating BC. It is reported that HER2 positivity is linked with resistance to endocrine therapy due to the inverse relationship between ER/PR and HER2 expression levels. For the HER2+ phenotypes, HER2 examination is established as a routine clinical practice before applying anti-HER2 therapy, including humanized monoclonal antibodies (mAbs) that bind to the extracellular domain of the HER2 receptor, etc., trastuzumab and pertuzumab, small-molecular inhibitors (lapatinib), and antibody–drug conjugates (ado-trastuzumab emtansine or T-DM1).

### 2.3. BRCA

Except for ER/PR−/HER2+ tumor subtypes, ER/PR/HER2− tumors, also called triple-negative breast cancer (TNBC), show aggressive behavior and poor outcome, with the highest mortality rate and recurrence, and are least likely to respond to hormone therapy. Despite the sensitivity to chemotherapy, the prognosis of conventional chemotherapy is still unsatisfactory. It was earlier reported that the somatic BC susceptibility gene (BRCA1/2) with a 12% mutation rate was found in 20% of TNBC patients, of which BRCA1 mutations are most likely to occur [7,8]. Mutation rates are higher in patients with BRCA1/2- multiple primary BCs than in single BC, indicating that the dysfunctional BRCA pathway in BC should be considered seriously [9]. The protein encoded by the BRCA gene participates in double-strand DNA break repairs through homologous recombination. Patients with advanced or recurrent metastatic BC with somatic mutations in the BRCA1/2 genes are more sensitive to DNA-damaging drugs, such as platinum drugs, or poly (ADP-ribose) polymerase (PARP) inhibitors, such as olaparib, due to homologous repair deficiency [10].

### 2.4. EGF

As a member of the EGF receptor family, which includes HER2, epidermal growth factor receptor (EGFR) activation has been shown to promote cell proliferation, motility, and survival through different signaling pathways via activation of various downstream signaling pathways [11]. Despite the fact that EGFR is frequently overexpressed in TNBC, tyrosine kinase inhibitors (TKIs), which are specific anti-EGFR agents, have only been used as part of the standard regimen for specific tumor types, such as gefitinib, erlotinib, afatinib, and osimertinib for non-small cell lung cancer (NSCLC), and erlotinib for pancreatic cancer [12]. TKIs approved for use in the clinic for BC are still being researched and developed [13].

### 2.5. VEGF

Vascular endothelial growth factor (VEGF) was identified and isolated as an endothelial cell-specific mitogen to induce physiological and pathological angiogenic processes, with consequent tumor cell dissemination [14,15]. Angiogenesis promotes vascularization, which means generation of new blood vessels by sprouting from existing blood vessels. Due to vascularization, BC cells overexpressing VEGF point to larger metastatic potential [16]. VEGF biomarkers were identified as one of the key players in BC liver metastases in a study based on bioinformatics and microarray gene expression analysis comparing data from liver aggressive and primary tumor specimens. It is also of interest that HER2+ BC can induce overexpression of VEGF, which may contribute to further cancer lethality through metastasis.

Although the magnitude of the effects of tumor heterogeneity on biomarker expression and its clinical significance remain uncertain, these biomarkers have been designed as special targets that tremendously support the principle of targeted therapy for BC.

## 3. Clinical Therapy

Before routine clinical therapy, imaging tests for BC detection are quite necessary. Different imaging modalities have been exploited to automate BC detection, such as mammograms, ultrasound, magnetic resonance imaging, and histopathological images based on molecular classification [17]. It is worth mentioning that, recently, researchers are focusing on a new area of artificial intelligence (AI)-based methods for BC detection, diagnosis, and treatment. With the help of robust AI algorithms and machine learning, early diagnosis and precise subtype classification become less time-consuming and more traceable, accurate, and convincing. According to Yusuf et al. [18], the model built based on the BreakHis dataset containing 7909 benign and malignant histopathological images collected from 82 patients could achieve more than 99% accuracy on image-level performance. Moreover, Alejandro et al. [19] verified that the evaluated AI system could obtain accurate BC detection, with an average 95% confidence interval [CI] of 0.840, which was higher than a breast radiologist. AI-based methods are, therefore, a promising modality for fully reliable and efficient BC predictions and diagnosis.

In the clinical treatment of BC, hormone (endocrine) therapy, radiotherapy, and chemotherapy are commonly used. Hormone therapy, whose indicators are ER and PR biomarkers, is widely and effectively used to treat ER+/PR−, ER−/PR+, and ER/PR+ BCs. According to the latest National Comprehensive Cancer Network (NCCN) guidelines for BC, premenopausal and postmenopausal patients are treated with tamoxifen (selective estrogen modulator, SERM) and tamoxifen or aromatase inhibitors (AIs), respectively. Although hormone therapy is beneficial for hormone-receptor-positive (HR+) patients, it also shows some drawbacks. According to the most recent NCCN guidelines, both tamoxifen and AI cause hot flashes and night sweats, as well as vaginal dryness. Furthermore, resistance to hormone therapies is a major issue that must be addressed [20]. Furthermore, as AI administration time is increased, bone-related adverse events, such as fractures, pain, and osteoporosis, increase significantly [21,22]. Acupuncture and the antidepressant venlafaxine, as well as the anticonvulsants gabapentin and pregabalin, have been shown in studies to be effective in treating hot flashes in BC survivors [23].

Because breast tissue is superficial and sensitive to radiotherapy, it is typically used as adjuvant therapy after BC surgery, such as breast-conserving surgery or radical mastectomy [24,25]. Radiotherapy reduces the risk of local recurrence of BC significantly [26], but it also causes radiation dermatitis, radiation pneumonia, bone marrow suppression, heart and lung injury, and radiation-induced malignancy [27,28,29].

Chemotherapy is used to treat advanced BC patients, which is important for preventing tumor recurrence and improving long-term survival [30]. Adjuvant chemotherapy and neoadjuvant chemotherapy are the two most common types. Adjuvant chemotherapy is chemotherapy administered after surgery to prevent distant metastasis or to postpone tumor metastasis and thus prolong patient survival. Neoadjuvant chemotherapy is chemotherapy administered before surgery that determines the tumor’s response to the treatment, allows for more breast-conserving surgery, and is widely used in both early and locally advanced breast cancer [31,32].

Chemotherapy drugs commonly used in clinical practice are divided into four types: anthracyclines, alkylators, antimetabolites, and taxanes. The heterogeneity of BC tumors causes tumor resistance to only one kind of chemotherapy drug. Therefore, a combination of multiple drugs is used to enhance the anti-tumor treatment effect. For example, the combination of cyclophosphamide, methotrexate, and 5-fluorouracil reduced the risk of recurrence by 30% in 10 years and 10-year overall mortality by 16% [33]. At the same time, a combination of targeted drugs, such as trastuzumab, has also achieved a better therapeutic effect on HER2+ BC [34]. However, resistance to chemotherapy drugs is the largest reason for their limited efficacy [35]. Meanwhile, cytotoxicity and adverse reactions to chemotherapy drugs still need to be considered in clinical application [36,37,38].

## 4. Targeted Therapy

### 4.1. Antibody Therapy and Inhibitors

#### 4.1.1. HER2 Targeted

In the past 20 years, two HER2-targeted mAbs have been approved for HER2+ BC under metastatic and adjuvant conditions: trastuzumab and pertuzumab. Trastuzumab, one of the first approved targeted anti-oncology therapies, is considered the basis for targeted treatment due to its effectiveness in women with BC harboring HER2 overexpression and/or amplification. Its anti-tumor action interferes with HER2 signaling following binding to the extracellular domain of the receptor through several mechanisms: antibody-dependent cellular cytotoxicity (ADCC), inhibition of the HER2 receptor dimerization, receptor internalization and/or degradation, and inhibition of downstream signaling pathways, i.e., the PI3K-AKT signaling pathway [39]. The overall survival (OS) improvements observed with trastuzumab were maintained after a median of more than 8 years of follow-up in a phase III study of patients with HER2-positive metastatic BC, resulting in an 8-year OS rate of 37% [40]. Trastuzumab has been assessed in other tumor types due to its high anti-tumor efficacy, including HER2-amplified gastric cancer [41] and biliary tract cancer [39]. Furthermore, current anti-HER2 mAbs treatments are more likely to be combined with chemotherapeutic drugs, such as trastuzumab deruxtecan (T-DXd) conjugated to an exatecan derivative (MAAA-1181a (DXd), which will be discussed in the following chapter regarding antibody–drug conjugates.Given that HER2 belongs to the receptor tyrosine kinase (RTK), except for anti-HER2 antibodies, drug development efforts have also been dedicated to targeting tyrosine kinases with small-molecule TKIs. Lapatinib, pyrotinib, neratinib, and tucatinib represent TKIs. TKIs inhibit kinase activity by penetrating membranes of tumor cells, thereby competing with ATP for the binding site of HER2 and finally blocking HER2 phosphorylation and downstream signaling transduction. Because of their low molecular weight, these HER-directed TKIs are more effective at penetrating the blood–brain barrier, making them more effective in patients with HER2 brain metastases and HER2-positive BC resistant to antibody therapy [42]. The ExteNET trial found that neratinib, compared to a placebo, improved invasive disease-free survival (iDFS) in patients with HER2/HR+ early-stage breast cancer after neoadjuvant/adjuvant trastuzumab-based therapy [43]. Except for that, the oral dosage takes appropriate palatability and swallowability into account, which have a significant impact on patient adherence.

#### 4.1.2. BRCA-Targeted

Based on significant PFS and OS benefits compared to standard chemotherapy, PARP inhibitors (PARPi) are clinically chosen in breast cancer settings. PARPi were discovered to have synthetic lethality with BRCA mutations. The main mechanism is to inhibit PARylation by binding to PARP and trapping inactive PARP on DNA, thereby blocking the replication forks, leading to their folding and the generation of double-strand breaks that finally cause tumor cell death [44]. The representative, olaparib, is approved in the USA for treatment of metastatic BC and in Europe for locally advanced or metastatic BC. For BRCA mutant cancer cells, the DNA is more easily damaged and needs to rely on Poly-ADP transferase to repair the DNA. Therefore, olaparib demonstrates its specific lethality by targeting BRCA mutant cancer cells other than healthy cells. However, the prospect of biomarker-targeted therapies for BC is provided by PARPi, and identifying patients who could potentially benefit from this treatment is noticeable. The BRCA gene test, an established predictive test in BC risk assessment, is recommended to guide therapeutic choice before and during the trial [45].

#### 4.1.3. EGFR-Targeted

Targeting EGFR with mAbs has been a successful strategy in different cancer subtypes, especially in NSCLC and colorectal cancer. Given the dramatic activity of these cancers and the overexpression of EGF biomarkers in up to 66% of TNBCs, efforts to target EGFR in BC have focused on TNBC. However, clinical efforts to target EGFR in TNBC have been unsuccessful until now, with increased toxicity and unsatisfactory clinical benefits [12]. Cetuximab is one of the candidates chosen primarily for clinical trials. In a phase II study designed for metastatic BC, Ixabepilone in combination with cetuximab did not show any signal of increased efficacy over single-agent chemotherapy [46]. The reason may be that few TNBCs are oncologically addicted to the EGFR signaling pathway, which renders them sensitive to EGFR inhibition [47].

#### 4.1.4. VEGFR-Targeted

As the first approved angiogenic mAb, bevacizumab (Avastin^®^) holds a pivotal position in targeted treatment of VEGF-A (referred to as VEGF) and even in targeted therapy. Marking the start of a new paradigm in oncotherapy, it remains the most broadly characterized anti-angiogenetic agent. Bevacizumab shows its potent inhibition of neo-vascularization by binding to soluble VEGF-A ligands in the circulation, preventing combination of VEGF-A with VEGF receptor (VEGFR), thereby activating the VEGF signaling pathways [48]. Therefore, in clinical regimens, angiogenesis-driven solid tumors, including TNBC [49] and HER2− BC [50], were focused on. Except for the strategy of using antibodies against VEGF isoforms, antibodies against VEGFR share the same anti-angiogenic potency. Ramucirumab is one of these candidates and occupies a niche in metastatic gastric cancer, NSCLC, and hepatocellular carcinoma (HCC) [51]. Until now, a few clinical trials have been designed to prove its potential efficacy in BC. A phase II study designed for unresectable, locally recurrent, or metastatic BC patients who had previously been treated with anthracycline and taxane therapy, unfortunately, failed to improve progression-free survival (PFS) and OS compared to eribulin monotherapy with ramucirumab in combination with eribulin, indicating that the efficacy and safety of ramucirumab in BC treatments still need to be evaluated [52].

### 4.2. Antibody–Drug Conjugates

The non-specific drug action of cytotoxic drugs is known as traditional chemotherapy regimens for cancer, which means rapidly dividing healthy and cancer cells are attacked, leading to side effects. By discovering subtle differences between cancer and normal cells, more effective and secure treatment options can be developed through targeted therapy research, one example of which is antibody–drug conjugates [53].

Antibody–drug conjugates (ADCs) are a new type of biological drug that consists of mAbs and small-molecule drugs coupled with bioactive linkers. Over the course of their development, two key factors that influence the effectiveness of ADCs have gradually emerged: the design of a proper linker between the mAb and the payload and the joining of a powerful cytotoxic agent to the mAb [54]. To improve these two factors, three generations of ADCs have been produced, which will be introduced in the next section with their specific design, mechanism of action, and therapeutic indications.

#### 4.2.1. The First-Generation ADCs

The first drug for ADCs is gemtuzumab ozogamicin (Mylotrag^®^), which is composed of gemtuzumab linked to N-acetyl-γ-calicheamicin dimethyl hydrazide via non-specific lysine conjugation and was approved in 2000 [55]. The linker contains a hydrazone bond that can be hydrolyzed in the acidic environment of the target cell, releasing the anti-tumor antibiotic calicheamicin.

Although this drug was designed to be hydrolyzed upon entry into the cell by the acidic environment of early nucleosomes and lysosomes, the linker was found to be unstable in the circulation, which led to premature release of toxic loads of calicheamicins that caused undesired toxicity, and, consequently, it was voluntarily withdrawn by Pfizer in 2010.

#### 4.2.2. The Second-Generation ADCs

The second-generation ADCs used different linkers to avoid premature drug release, as observed in Mylotrag^®^. For example, the T-DM1 ADCs (Kadcyla^®^), approved by the FDA in 2013 for treatment of HER2− BC, are composed of anti-HER2 IgG1 trastuzumab linked to DM1 through a nonreducible heterobifunctional thioether linkage containing an N-hydroxysuccinimide ester (SMCC), which was found to be less toxic, more efficacious, and pharmacokinetically stable [56].

However, one study found that Kadcyla^®^ increased the risk of radiation necrosis, which may be associated with its attribution to normal glial cell death and dysfunction [57].

#### 4.2.3. The Third-Generation ADCs

The third-generation ADCs are characterized by moderately stable linkers with short half-lives and bystander effects, followed by selective binding to antibodies with high drug-to-antibody ratios (DAR) and utilization of drugs with nanomolar toxicity on DNA targets [58].

An example is sacituzumab govitecan (Trodelvy^®^), which consists of a monoclonal antibody hRS7 lgG1Κ [anti-trophoblast cell-surface antigen 2 (TROP-2)] linked to an active metabolite of irinotecan called SN-38 via a hydrolyzable CL2A linker. At the same time, pH-sensitive benzyl carbonate bonds were used to release SN-38 under acidic conditions on target cells and their tumor microenvironment (TME) to reduce off-target toxicity, and studies have demonstrated that Trodelvy^®^ has a high DAR (~8: 1) [59,60].

## 5. Targeting Drug Delivery Systems

Nanoparticles (NPs) are stable colloids made up of polymeric materials with varying properties, such as polymers, lipids, or metals, with particle sizes smaller than 1000 nm. Drugs are typically covalently attached or encapsulated on the surface of NPs and passively or actively targeted to cancer cells, which not only increases their solubility but also extends their half-life and improves bioavailability via the tumor tissues’ enhanced permeability and retention effect (EPR) properties [61,62]. Because NPs are smaller than the cut-off size of tumor vascular pores, they can cross the cell membrane and preferentially accumulate in tumor cells. Until now, several NPs-related products have been approved or currently evaluated in clinical trials for BC (Table 1), indicating that NPs have promising development prospects as drug carriers. Nowadays, a wide range of materials with varying properties are used as NPs, with varying particle sizes, shapes, surface chemistry, and residence times in circulation (Figure 2), all of which can affect their effectiveness [63]. Under these conditions, it is critical to choose materials that are compatible with the properties of the encapsulated drug.

### 5.1. Inorganic NPs

Inorganic NPs are developed using inorganic elements, including metals, which are one of the most typical elements that are widely utilized in diagnosis and treatment of cancer, such as magnetic resonance imaging (MRI) or positron emission tomography (PET). These NPs are capable of controlling the release rate of drugs via different stimuli, increasing drug targeting and reducing adverse effects. For BC, inorganic NPs focus on gold and magnetic NPs, which have diverse molecular mechanisms for inducing apoptosis in tumor cells due to their unique magnetic, optical, or thermodynamic properties, such as conversion of electromagnetic radiation into heat, absorption of near-infrared light, plasma resonance, etc. [64,65,66].

#### 5.1.1. Gold NPs

When internalized, gold nanoparticles (GNPs) have been investigated as radiosensitizers due to their chemical stability and unique property of causing enhanced radio-sensitivity of cells [67]. GNPs have a “photothermal effect” that involves conversion of light energy into kinetic energy after absorption of photons, with some of the latter eventually expressed in the form of heat. GNPs have also been used as photothermal agents to inhibit tumor growth by generating heat upon irradiation with near-infrared light since tumor cells are less heat-resistant than normal cells [68].

The positively charged Au^+^ ions in GNPs function as tumor-targeting carriers by attracting negatively charged biomolecules and forming stable chemical bonds. Meanwhile, GNPs are linked to functional organic ligands or polymers via thiol- or nitrogen (N)-containing linker molecules, resulting in different physiological effects [69]. For example, Li et al. [70] used a calcium phosphate (CaP) shell and a removable gold nanorod yolk to form CaP-based yolk–shell NPs. The loading efficiency of the doxorubicin (DOX) molecules could reach up to 100%. Moreover, it has pH/NIR dual-response capability to accumulate and induce DOX release in tumors under acidic environments or near-infrared laser stimulation.

#### 5.1.2. Magnetic NPs

Magnetic NPs have superparamagnetic features in their crystalline core that allow their microwave magnetic response to be modified under an external polarized magnetic field without affecting the surrounding environment, allowing them to be used as a contrast agent in cancer diagnosis and tumor imaging. Magnetic NPs also have the ability to function properly and selectively target tumor tissues with tumor-cell-receptor-specific antigens, making them promising carriers for targeted drugs in research [71].

Zou et al. [72] developed mesoporous magnetic NPs loaded with DOX and coated with chitosan to improve their biological properties. These NPs showed high DOX loading capacity and the potential to target BC under alternating current (AC) electromagnetic fields. Apart from that, according to another team [73], polyethylene-glycolized magnetic NPs coupled with anti-VEGF antibodies were used for DOX delivery, which enabled these NPs to accumulate at the tumor site while their magnetic cores provided strong signals that were detected by MRI for real-time monitoring.

### 5.2. Polymeric NPs

The diameter range of polymeric NPs can be controlled within 10–1000 nm, which enables extended blood circulation of the encapsulated drugs through the EPR effect. As an ideal drug delivery system, polymeric NPs are available to target and control the release of drugs by modulating the properties of the polymer or modifying the surface with various ligands to improve the bioavailability and therapeutic index [74].

#### 5.2.1. Polymeric Micelles

Micelles are colloidal dispersions of nanoparticles formed by self-assembled amphiphilic copolymers in specific media with distinct hydrophilic and hydrophobic blocks. The majority of hydrophilic blocks are composed of polyethylene glycol (PEG), while hydrophobic blocks are made of poly(lactide) (PLA), poly(ethylene oxide) (PEO), or poly(ε-caprolactone) (PCL), with polymer shapes, including spheres, layers, and rods [75,76].

Polymeric micelles are not only easily prepared, more stable, and biocompatible but also have lower critical micelle concentrations than previous surfactants that were commonly used to solubilize insoluble drugs. The stimuli-responsive cleavable bonds or targeted ligands on the blocks of polymeric micelles enable micelles to respond to various stimulating factors of the microenvironment, such as temperature, pH, reducing agents, specific enzymes, or near-infrared irradiation (NIR). These characteristics enable stimuli-responsive micelles to be widely used in precisely targeted delivery of drugs (Table 2).

Peng et al. [77] constructed Herceptin-conjugated PCL-PEG worm-like nanocrystal micelles for HER2+ breast cancer using paclitaxel (PTX) and Herceptin and found them stable in blood circulation and TME with specific HER2+ tumor cells targeting. Garg et al. [78] formed PEO-poly(α-benzyl carboxylate-ε-caprolactone) (PEO-PBCL) by introducing pendant benzyl carboxylate groups to the PCL segment of PEO-PCL and the core-forming block coupled to the NIR probe Cy5.5 to develop traceable polymeric micelles. Subsequent tracer studies showed that these modified micelles had higher accumulation at tumor sites, were more stable, and could track disease progression in real-time with in situ BC mouse tumors.

By introducing ionizable groups into the copolymer, micelles can be released in the acidic pH of the TME. Emami et al. [79] used α-tocopherol as a hydrophobic segment and electrically coupled it to a pH-cleavable cis-aconitic anhydride. This conjugate then reacted with heparin via a steroid bond to form micelles that encapsulated the anti-tumor drug docetaxel in aqueous media (DTX). These experiments demonstrated that the micelles improved not only the chemotherapeutic drug solubility and specific distribution but also had desirable biocompatibility, high pH sensitivity, and effectively prolonged blood circulation time.

**Table 2 cancers-14-05456-t002:** Examples of stimuli-responsive polymeric micelles in breast cancer therapy.

Stimulus	Assembly Units	Payloads	Characteristics	Reference
Temperature	PNIPAM-co-DMAAm-PLA	Shikonin	Thermo-responsive deformation of micellar structure	[80]
	mPEG-PDLLA	T1^+^PS	Thermo-responsive forming hydrogel	[81]
pH	PF127-PMVEMA	DOX	pH-responsive dissociation	[82]
	Dextran-retinal	DOX	pH-responsive deformation of micellar structure	[83]
	mPEG-b-PDPA	SCB	pH-responsive deformation of micellar structure	[84]
	mPEG-PBAE	urushiol	pH-responsive deformation of micellar structure	[85]
Enzyme	mPEG-S-S-VES	DTX	MMPs and GSH selective cleavage	[86]
Light	PEG-IR780-BIIB021	IR780	Photo-responsive fluorescence	[87]
Redox	F127-SS-TOC	/	Redox-sensitive disulfide bonds as linkers	[88]
Magnetic	Soluplus^®^	DCT^+^MNPs	magnetic response	[89]
pH, NIR, temperature,	mPEG-PAAV	DOX^+^IR780	pH-responsive upper critical solution temperature and NIR absorber	[90]
pH, temperature	PHEMA-g- (PCL-BM: beta-CD-star-PMAA-b-PNIPAM)	DOX	pH- and thermo-responsive dissociation	[91]

#### 5.2.2. Nanocapsules

Nanocapsules are framed with a vesicular structure in which the drug is not only embedded in a cavity surrounded by a polymeric membrane but also contains liquid or solid active substances [92]. They consist of a polymeric shell layer and an oil core, which increases the aqueous solubility of lipophilic drugs.

Acting as a targeted drug carrier, polymeric nanocapsules could prolong the drug’s circulation time, delay, or control the drug’s release and also improve its efficacy [93]. In a study of three different lipid nanosystems, the researchers found that, when DOX was enclosed in lipid nanocapsules, the half-maximal inhibitory concentration (IC50) decreased, implying less drug dosage and toxicity [94]. Apart from that, one group of researchers synthesized nanocapsules consisting of hyaluronic acid (HA) and hydroxychloroquine (HCQ) at pH 7.4, which is pH- and redox-dual-responsive [95]. Through experiments, they demonstrated that these nanocapsules have remarkable targeting and selectivity for 4T1 cells.

#### 5.2.3. Nanospheres

Nanospheres are presented as homogeneously dispersed matrix structures that are prepared from biodegradable polymers and serve as fluorescent nanoprobes for tumor cell imaging [96]. Wu et al. [97] developed a method to produce magnetic and fluorescent nanospheres for magnetic capture and fluorescent labeling of circulating tumor cells, which can be applied for efficient detection and isolation of tumor cells.

In a targeted drug delivery system, drug molecules are trapped in the nanospheres to avoid drug degradation by enzymes or other substances in circulation and improve the efficacy of drugs [98]. Tang et al. [99] fabricated self-assembled mRNA nanospheres loaded with DOX, and the experiments showed that the nanospheres efficiently expressed apoptogens and increased the necrosis of tumor tissues, exhibiting the synergistic effects of gene chemotherapy. Moreover, Vahab and Alireza developed nanospheres that were composed of folic acid and poly(methacrylic acid) with pH responsiveness [100]. With the attachment of folic acid to the surface, these nanospheres targeted folate receptors overexpressing BC cells to achieve targeted delivery of toxic loads.

#### 5.2.4. Dendrimers

Dendrimers are nano-sized-scale polymers composed of monodisperse molecules. As targeted drug carriers, dendrimers are presented as derivatized branched structures containing a variety of optional modifications for high drug loading capacity and targeted drug delivery [101]. Anti-tumor drugs are packaged in the lumen of the dendrimers to improve their stability and reduce immunogenicity. Sergio et al. [102] coupled D-glucose to methotrexate-loaded polyamidoamine (PAMAM) to obtain glycosylated dendrimers. The molecule was proven to improve cellular targeting, and the cytotoxicity was higher than free methotrexate.

In addition, Aleanizy et al. [103] formulated a PAMAM 4th generation (G4) dendrimer presented with trastuzumab to deliver an adjuvant. They demonstrated that these dendrimers were more selective, cytotoxic, and had higher cellular uptake than the free drugs, which were shown to be promising anti-breast-cancer drug-targeted delivery systems.

### 5.3. Lipid NPs

Lipid NPs contain aqueous, oily, or solid nuclei and are surrounded by lipid layers [104]. They are widely used in development of drug formulations, especially in the field of nucleic acid drug delivery, as they have numerous advantages, including stable drug loading, reduced off-target side effects, and improved delivery efficiency. Lipid NPs are biocompatible, and, more importantly, they can increase stability by adding cholesterol or modifying the surface with PEG. Additionally, other surface modifications, such as peptides, antibodies, and small molecules, allow targeted delivery.

#### 5.3.1. Solid Lipid NPs

Solid lipid NPs (SLNs) consist of natural or synthetic solid lipid carriers in which drugs are embedded in the lipid core or bound to the lipid surface. They are more stable than polymeric materials, allowing sustained drug release, thereby reducing the dosage, decreasing toxicity, and delaying the onset of drug resistance. For example, in one study, SLNs loaded with DTX were found to have a higher tolerated dose and lower organ toxicity in mice compared to paclitaxel [105]. More importantly, Venkata et al. [106] modified SLNs with receptors for advanced glycation end-products (RAGE) antibodies to deliver diallyl disulfide, which was site-specific compared to the SLNs without antibodies in MDA-MB231 cells overexpressing RAGE. Therefore, these SLNs overcome the off-target effects of cytotoxic agents and multidrug resistance via drug efflux transporters. In addition, increased programmed apoptosis was found in SLNs loaded with tamoxifen due to mRNA and miRNA expression profiles that control apoptosis, but it still needs to be verified through further experiments [107].

#### 5.3.2. Liposomes

Liposomes are vesicles composed of degradable and biocompatible lipid bilayers, which allow them to store hydrophilic drugs internally while hydrophobic drugs remain preserved in the bilayer [108]. Liposomes are highly appealing as carrier systems for targeted therapies due to their ease of production and modification, which can be used to achieve specific biological effects and create active targeting in the TME through a variety of modifications. Liposomes improve drug delivery by increasing uptake and accumulation in tumor tissues, eliminating off-target toxicity and reducing side effects. These modifications in structures and surfaces should focus on designing a co-localization system between the drug and BC cells and enhancing the triggered release of the drug in the TME [109].

Liang et al. [110] used peptide-p37-modified cationic liposomes CDO14 to deliver survivin siRNA for treatment of heat-shock-protein-gp96-overexpressed breast cancer. Peptide p37 is an inhibitor of gp96, which is a new target for tumor therapy and can enhance targeting after modification of liposomes. Experiments have demonstrated high efficiency of p37-CDO14 gene silencing, which significantly enhanced the anti-tumor effect compared to the unmodified liposomes. One group of researchers applied thermosensitive liposomes encapsulated with the photosensitizer indocyanine green and the anti-tumor natural drug parthenolide for synergistic treatment of TNBC [111]. Under NIR, indocyanine green released heat to change the structure of thermosensitive liposomes and unleash the drug. The tumor suppression rate of this liposome was 2.08-fold higher than that of paclitaxel, but it still needs further research and validation for in vivo evidence.

Jain et al. [112] developed a pH-sensitive liposome-loaded DTX with surface-coupled VEGF antibody for BC treatment. It has been indicated that there was greater cellular uptake, a higher percentage of drug release in an acidic environment, and a longer half-life with the drug delivery system compared to the free DTX. Cao et al. [113] used a biomimetic drug delivery strategy to construct pH-sensitive liposomes coated with macrophage membranes to deliver the cytotoxic anticancer drug emtansine to enhance the specific metastatic targeting ability of liposomes. The liposomes were proven to improve the specific targeting ability for lung metastases of BC, thereby inhibiting it significantly.

## 6. Immunotherapy

In contrast to chemotherapy, which utilizes its toxicity to directly kill cancer cells, immunotherapy is based on the principle of controlling the signals of cell growth and enhancing or stimulating the natural immune response to fight against the cancer cells. Therefore, continuous treatment could be avoided to reduce side effects by reprogramming the immune system. However, the majority of tumors possess immune escape capabilities via decreased expression of neoantigens, downregulated levels of antigen-presenting cells, increased expression of anti-apoptotic proteins, and release of inhibitory cytokines, such as transforming growth factor-beta (TGF-β), interleukin-10 (IL-10), and programmed death-ligand 1 (PD-L1). Therefore, the key to immunotherapy is directing immune cells to specifically recognize cancer cells and breaking the tolerance to trigger autoimmunity. Although breast cancer has not been classified as immunogenic, there are numerous immunotherapeutic agents that modulate its interaction with the immune system, which are discussed in detail in the sections that follow [114].

### 6.1. Checkpoint Inhibitors

Activation of T cells necessitates not only the first signal provided by antigenic stimulation but also the second signal from co-stimulation and cytokines. Therefore, immune checkpoints function as suppressors in the immune system, modulate the immune response, maintain self-tolerance, avoid excessive activation of the system, and prevent damage to the autologous tissues [115]. This dual recognition is essential to maintain immune homeostasis and is also utilized by cancer cells to suppress anti-tumor T cell responses while avoiding abnormal activation of the immune system. The immune checkpoint inhibitors (ICIs) were exploited to develop drugs that could reactivate the immune system for antigen presentation and kill tumor cells.

Programmed death-1 (PD-1) and cytotoxic T lymphocyte antigen 4 (CTLA-4) are the first checkpoint receptors to offer a wide range of prospects for immunotherapy. However, they differ in the mechanism of attenuating T cell activation. CTLA-4 is upregulated after T-cell receptor (TCR) ligation and inhibits the activation by attenuating the positive co-stimulation of CD28, while PD-1 is induced upon the activation to attenuate TCR signaling and inhibit T cell proliferation and survival by recruiting tyrosine phosphatases upon binding to its ligands PD-L1 or PD-L2 [116,117]. These two representative immune checkpoints are described further below.

#### 6.1.1. PD-1/PD-L1

PD-1 is a co-inhibitory receptor that regulates T cell effector function, which means blocking this pathway can enhance the immunogenicity of T cells and promote anti-tumor effects. Although the major ligands of PD-1 are expressed in tumor-associated antigen presentation cells (APCs), PD-L1 is present in a variety of tumor cells, while PD-L2 is predominantly found in hematologic cancer cells [118]. Therefore, in most studies, PD-L1 is considered a potential response marker for checkpoint therapies.

As for breast cancer, PD-L1 protein is expressed in 20–30% of patients, with TNBC exhibiting the highest constitutive expression, making this receptor a promising area for anti-tumor research [119]. For example, a study demonstrated that atezolizumab, an FDA-approved anti-PD-L1 antibody, enhanced the cytotoxicity and apoptosis mediated by T cells in TNBC with high PD-L1 expression [120]. In addition, another clinical phase III study that used atezolizumab with paclitaxel for treatment of patients with metastatic TNBC found this combination to significantly prolong PFS compared to the control group (NCT02425891).

However, some researchers pointed out that almost all anti-PD-L1 antibodies on the market have serious side effects due to their high immunogenicity [121,122]. Therefore, Ma et al. [123] screened a human-derived protein scaffold, U1, small nuclear ribonucleoprotein polypeptide A (snRNPA), which is complementary in shape to the domain of the PD-L1 binding receptor, by constructing a library and found that this combination inhibited PD-1/PD-L1 interaction. Compared to antibodies, the human-derived protein scaffold has a longer half-life and better permeability because of its lower molecular weight and immunogenicity since this protein scaffold exerts anti-tumor activity by reactivating tumor-suppressed T cells and is unable to coordinate direct cytocidal effects.

#### 6.1.2. CTLA-4/CD28

CTLA-4 is a fundamental immunoregulatory molecule that raises the activation threshold of T cells and attenuates the anti-tumor response, which is expressed on the surface of T cells and T regulatory cells (Tregs). However, CTLA-4 has the same functional expression on tumor and T cells. For example, its expression on BC cells inhibited maturation of dendritic cells, which is CTLA-4 dependent. Meanwhile, another study demonstrated that TNBC has the highest CTLA-4 expression of all types of BC [124,125].

There are two categories of monoclonal antibodies targeting CTLA-4, ipilimumab, and tremelimumab in BC. For example, McArthur et al. [126] evaluated the efficacy of cryoablation and ipilimumab prior to BC surgery. Clinical studies have found favorable immune effects in the combination of these two approaches through cryoablation-mediated tumor antigen presentation and ipilimumab-monoclonal-antibody-mediated immunomodulation, resulting in a synergistic anti-tumor immune response.

Aside from that, a clinical phase II trial combining the anti-PD-1 monoclonal antibody nivolumab with ipilimumab in metastatic hypermutated HER2-negative BC is currently underway (NCT03789110). Furthermore, a clinical phase II trial validated the efficacy of PD-L1 inhibitor durvalumab in combination with tremelimumab in patients with metastatic ER+ or TNBC, finding that TNBC had a higher response rate and clinical benefit to immunotherapy [127].

### 6.2. CAR T Cell Therapy

Antibody-derived chimeric antigen receptor (CAR) T cell therapy, known as adoptive cell therapy, involves genetic modification of host cells to express anti-tumor T cell receptors or chimeric antigen receptors, resulting in anti-tumor responsiveness. CAR is a synthetic receptor that consists of extracellular single-chain variable fragments, transmembrane domains, immunoreceptor tyrosine-based activation motifs, and co-stimulatory signals [128,129,130]. It evolved in the fourth generation as cytotoxicity was reduced and specific co-stimulatory structural domains were added. The fourth generation CAR T cell added IL-12 to eliminate antigen-negative cancer cells based on insertion of CD28 to improve T cell activity in the second generation compared to the first generation, which had only a single activation structural domain resulting in cytotoxicity [131,132].

For BC, it is essential to select the optimal tumor-specific or tumor-associated antigens to enhance tumor auto-immunogenicity and succeed in adoptive T cell therapy. For example, Zhou et al. [133] selected mucin 1 (MUC1), which is expressed in more than 90% of BCs and 95% of TNBC, as the target tumor antigen. They transduced human T cells with chimeric antigen receptor MUC28z, which recognizes the aberrantly glycosylated MUC1 in all BC subtypes. This therapy has target-specific cytotoxicity and significantly reduced the growth of TNBC in experimental tumor xenograft models. Moreover, since EGFR is highly expressed in TNBC, another team chose EGFR as a candidate to design CAR T-cell therapy. It was shown to inhibit TNBC with limited toxicity to normal tissues both in vitro and in vivo [134].

### 6.3. Cancer Vaccines

With the development of cancer immunotherapies, such as immune checkpoint inhibitors and adoptive cell transfer, the necessity for pre-sensitized tumor antigens, activating long-term immunological memory, and amplifying the number of tumor-reactive T cells in the naive repertoire gradually arose, and, consequently, the emergence of cancer vaccines occurred. Among immunotherapeutic strategies, tumor vaccines are primarily active immune approaches, which enhance anti-tumor immune responses via carrying tumor antigens that activate autologous immune cells in the patient. To date, the most widely used tumor vaccines are peptides derived from tumor antigens, while other types include dendritic cell (DC) and DNA-based vaccines [135,136,137].

Peptide-based vaccines possess the advantages of being easy to synthesize, cost-efficient, and having acceptable side effects. For example, Burn et al. [138] tested glycolipid-peptide conjugate vaccines to activate cytotoxic T-cell responses, which delayed breast tumor growth and impeded tumor lung metastasis in an experimental metastasis model. Meanwhile, the efficacy of these vaccines was also demonstrated in BALB/cJ mice, which have smaller and more T helper 2 (Th2)-skewed natural killer T (NKT) cell populations.

In addition, since DCs process and present antigens to T cells, the researchers isolated immature DCs from the peripheral blood of cancer patients and stimulated them with compatible antigens and cytokines. Afterward, the mature DCs were reinfused to launch robust anti-tumor immune responses. Tomasicchio et al. [139] developed a DC vaccine that proved to be cytotoxic to autologous BC cells in vitro and was shown to elicit a vigorous, dose-dependent, and antigen-specific cytotoxic T-lymphocyte response.

Moreover, another team produced a DNA vaccine that utilizes the specific target mammaglobin-A (MAM-A), a gene that is expressed only in BC and overexpressed in 40–80% of this cancer [140]. It was demonstrated in a phase I clinical trial that the number and frequency of specific CD8+ T cells significantly increased in patients, with preliminary evidence of improved PFS.

### 6.4. Tumor-Microenvironment-Targeted Therapy

The tumor microenvironment (TME) of BC is intricately composed of vascular stromal cells, fibroblast cells, extracellular matrix (ECM), and various types of innate and adaptive immune cells (Figure 3), together with multiple extracellular soluble molecules, such as cytokines, chemotactic and growth factors [141].

Although the search has been pursued to identify genetic mutations that drive cancer to develop, recent studies have revealed that TME plays a crucial regulatory role in cancer progression and its immune escape. For example, fibroblasts in normal tissues facilitate repair and regeneration, while they can be activated into cancer-associated fibroblasts (CAFs), which produce chemokines to promote inflammation and fibrosis of the tumor [142]. Therefore, CAFs have become a popular target for BC research. For instance, a study by Yao et al. [143] demonstrated that artesunate and dihydroartemisinin derivatives of artemisinin inactivated CAFs and reduced the interaction between tumor and TME by inhibiting the TGF-β signaling pathway. In addition, another team identified an aberrantly expressed miRNA, LNA-i-miR-221 (miR-221), which initiated proliferative and migratory effects in BC via interfering with the A20/c-Rel/CTGF signaling pathway in CAFs [144]. Therefore, they synthesized locked nucleic acid inhibitors of miR-221 to block these pro-tumor effects.

Meanwhile, macrophages and other leukocytes are recruited for immunosuppression and the spread of cancer cells. Tumor-associated macrophages (TAMs) account for more than 50% of the tumor mass and have two general phenotypes, the M1 phenotype, which activates anti-tumor immunity, and the M2 phenotype, which promotes angiogenesis. Although these two phenotypes depend on the stage of tumor progression, TAMs in BC are typical of M2 macrophages [145,146]. Tan et al. [147] targeted the lysine-specific demethylase 1-nuclear REST corepressor 1 (LSD1-CoREST) complex, which was differentially expressed between M1 and M2 macrophages in TNBC and converted the tumor-promoting M2 macrophages into the less aggressive M1 phenotype. However, the related mechanism and implications require further investigation.

## 7. Conclusions

In the face of BC, a refractory disease, clinicians no longer have only two means, surgery and radiation therapy. As BC research has gradually entered into a molecular-biology-driven research phase, biomarkers routinely selected for histopathology examinations cannot fully satisfy the demand for precise detection. As a result, scholars are motivated to detect more specific markers and refine molecular subtypes with state-of-the-art technologies, including AI-based methods. Obviously, all the efforts paid to molecular classification would expedite precise detection and treatment of heterogeneous BC. For biomarkers, molecular-targeted therapy with mAbs, inhibitors, chemotherapeutic drugs, or any combination of them is proven effective in clinics, with the improved indicator of OS and PFS. Despite the advantages of high adherence by oral inhibitors and high specificity and affinity to a wide variety of molecules by mAbs, inadequate pharmacokinetics and tissue accessibility and impaired interactions with the immune system are the limitations of those therapies, especially mAbs treatments, which need to be conquered. Targeted drug delivery systems are thus selected as carriers to alleviate targeted therapy deficiencies. Due to the alternative active or passive targeting capability and the unique size effect of NPs, it is a more straightforward mission to achieve remarkable aggregation on the targeted lesion site, improve treatment effects, and reduce the toxicity or side effects. Therefore, a research focus for precise and efficient BC therapies is merging the delivery system and targeted drugs joining forces.

Moreover, with enhanced awareness of the importance of the immune system during tumorigenesis, stimulating the natural immune response to fight against cancer cells shows its efficacy. Owing to the heterogeneity, various immune cells, fibroblasts, and the extracellular matrix located in the TME interact with each other to promote proliferation and migration of the tumor and inhibit the normal function of immune cells in the tumor. Specifically, immune cells change from “soldiers” into “enemies” for the host, such as polarization of M1-like to M2-like TAMs. The interaction of the tumor cells and TAMs promotes rapid growth of the tumor, inhibits immune function, and leads to treatment failure. Given the difficulties mentioned above, an efficient combinatorial regimen of targeted therapy and immunotherapy may help to cure BC. One option is to multi-target different subtypes of tumor cells and M2-like TAMs. On the other hand, while BC is classified as a “cold tumor”, tumor-specific antigens delivered with CAR T therapy and cancer vaccines have the possibility to enhance tumor auto-immunogenicity and succeed in adoptive T cell therapy. The rise of immune checkpoint inhibitors, genetic immune therapy, and tumor-microenvironment-targeted therapy, all of which have the potential to become powerful armaments, is also hastening innovative treatment of BC. While a complete cure for BC may be challenging to achieve, new applications of targeted therapy and immunotherapy offer hope.

## Figures and Tables

**Figure 1 cancers-14-05456-f001:**
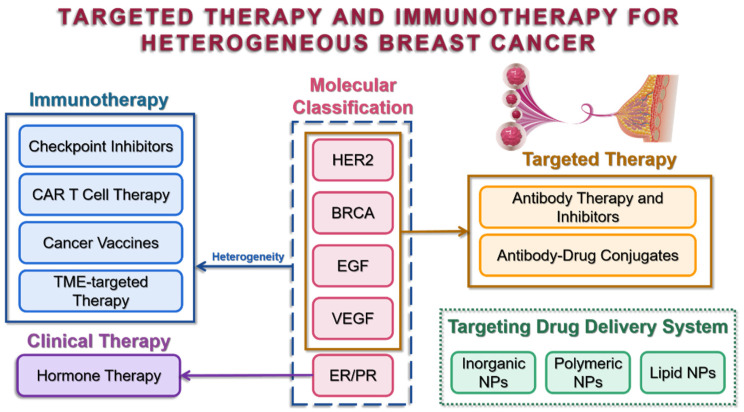
Molecular classification of breast cancer and corresponding treatments.

**Figure 2 cancers-14-05456-f002:**
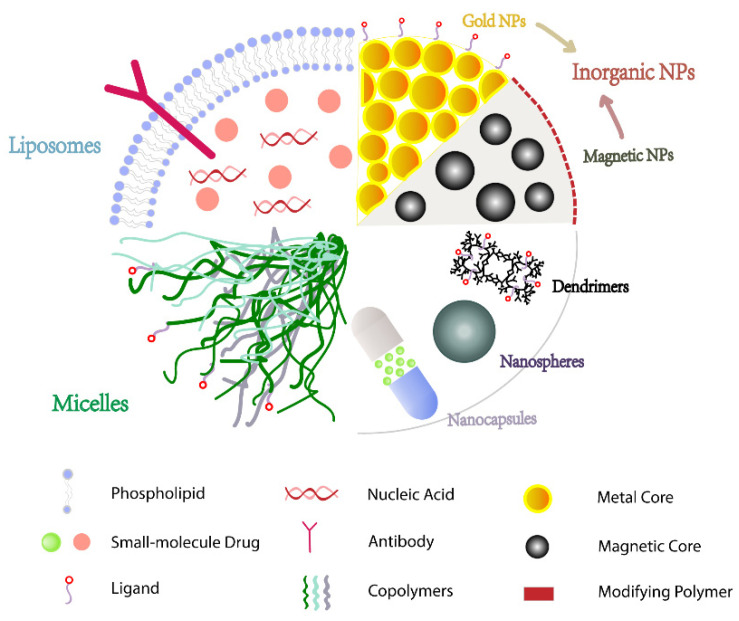
Nanoparticles with varying characteristics in targeting drug delivery systems.

**Figure 3 cancers-14-05456-f003:**
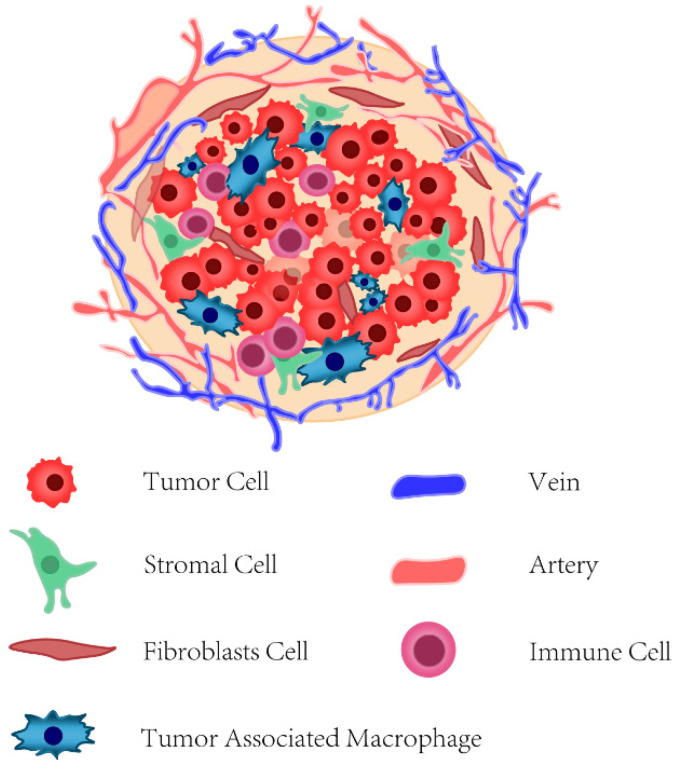
The tumor microenvironment of breast cancer.

**Table 1 cancers-14-05456-t001:** Nanoparticles approved or currently in clinical trials for breast cancer.

Delivery System	Drugs	Status
Liposomes	Doxil/Calyx	Approved for marketing in 1995
	Myocet	approved for marketing in 2001
	Trastuzumab, non-pegylated liposomal doxorubicin	Phase I
	Mitoxantrone	Phase II
	Irinotecan	Phase I
	TheromDox	Phase II
	DepoCyte	Phase III
	Liposomal annamycin	Phase I & II
	Liposomal Irinotecan, Pembrolizumab	Phase II
Polymeric	Abraxane	Approved for marketing in 2005
	Genexol-PM	Approved for marketing in 2006
	NK-105	Phase III
	Nanoxel	Phase I
	BIND-014	Phase I

## Data Availability

Not applicable.

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
