# Peer review of "Targeted Therapy and Immunotherapy for Heterogeneous Breast Cancer"

_cancers, 2022, doi:10.3390/cancers14215456_

Round 1
Reviewer 1 Report
This review article by Sun et al. is informative and comprehensive. The Authors concisely present a lot of information and provide insights regarding the classifications of heterogeneous breast cancer, as well as targeted therapy and immunotherapy. They also provide details about targeted drug delivery systems. There are a few aspects in which the manuscript could be improved:
1. The Conclusion section seems a bit lacking in content. The manuscript includes various information and covers several clinical challenges regarding breast cancer, so there are various points that the Authors could discuss in more detail. Perhaps an additional section could be added, titled “Discussion” or “Future perspectives”, for example, where the Authors provide more in-depth critical commentary, their take on which area is the most promising to advance into clinical practice, challenges that we still have to overcome, novelties in research that are promising but not thoroughly examined yet, etc.
2. The manuscript would greatly benefit from the addition of some Figures. While the Tables are helpful for the reader, the visualization of some of the information provided would enhance the quality of the manuscript.
Author Response
Please see the attachment.
Dear Editor and reviewers,
We thank the reviewer for the valuable comments and the time invested in optimizing the manuscript.
We have now submitted the revised manuscript. We have made extensive modifications to our manuscript and highlighted them in our revised manuscript. The “response to reviewer comment” is listed below. The revised submission has been fully proofread, agreed upon by all authors, and is considered final as there will be no further changes allowed should the manuscript be accepted.
Thank you very much for considering our manuscript for publication, and I am looking forward to hearing from you soon.
Sincerely yours,
Xiaolu Sun
School of Pharmacy
Shanghai Jiao Tong University,
Shanghai 200240, China.
E-mail: sxl_ruby@sjtu.edu.cn

Reviewer 2 Report
This manuscript present the review of wide range of molecular classifications of heterogeneous breast cancer (BC), emphasize targeted therapy and immunotherapy, and provide insights into the significance of targeted drug delivery systems. Below are my comments:
1. The authors must show the graphical/schematic roadmap for the study.
2. To make a complete REVIEW paper. The authors also provide the BC detection methods in detail, including recent Artificial Intelligence based methods.
3. It would be interesting to see the figures related to drug delivery systems (wherever appropriate) for the given methods.
Thus, recommend a Major Revision
Author Response

(The authors gave the same response as above.)

Reviewer 3 Report
In the manuscript entitled “Targeted Therapy and Immunotherapy for Heterogeneous Breast Cancer” by Xiaolu Sun et al, the authors provide a review of literature focused on molecular classifications of heterogeneous BC, targeted therapy and immunotherapy.
The manuscript is well composed with clear organization and the conclusion is valuable for future clinical research. The article is interesting and could be considered for publication. However, there are still some aspects should be optimized.
Some of the problems are summarized below:
1. For a smoother reading, I advocate that the authors merge Section 4.1.1. and 4.1.2.
2. The author should add a whole new section for summarize the benefits and drawbacks of targeted therapy and immunotherapy in BC.
Author Response

(The authors gave the same response as above.)

Round 2
Reviewer 2 Report
Accept in the present form.